# Dose-Ranging Plasma and Genital Tissue Pharmacokinetics and Biodegradation of Ultra-Long-Acting Cabotegravir In Situ Forming Implant

**DOI:** 10.3390/pharmaceutics15051487

**Published:** 2023-05-13

**Authors:** Isabella C. Young, Allison L. Thorson, Roopali Shrivastava, Craig Sykes, Amanda P. Schauer, Mackenzie L. Cottrell, Angela D. M. Kashuba, Soumya Rahima Benhabbour

**Affiliations:** 1Division of Pharmacoengineering and Molecular Pharmaceutics, UNC Eshelman School of Pharmacy, University of North Carolina at Chapel Hill, Chapel Hill, NC 27599, USA; 2Joint Department of Biomedical Engineering, North Carolina State University and The University of North Carolina at Chapel Hill, Chapel Hill, NC 27599, USA; 3Division of Pharmacotherapy and Experimental Therapeutics, UNC Eshelman School of Pharmacy, University of North Carolina at Chapel Hill, Chapel Hill, NC 27599, USA

**Keywords:** biodegradable, long acting, HIV, injectable, pharmacokinetics, polymer

## Abstract

HIV continues to affect millions of men and women worldwide. The development of long-acting injectables for HIV prevention can overcome adherence challenges with daily oral prevention regimens by reducing dosing frequency and stigma. We previously developed an ultra-long-acting injectable, biodegradable, and removeable in situ forming implant (ISFI) with cabotegravir (CAB) that demonstrated protection after multiple rectal SHIV challenges in female macaques. Here, we sought to further characterize CAB ISFI pharmacokinetics (PK) in mice by assessing the effect of dose and number of injections on CAB PK, time to completion of CAB release and polymer degradation, long-term genital tissue PK, and CAB PK tail after implant removal. CAB concentrations in plasma were above the benchmark for protection for 11–12 months with proportionality between dose and drug exposure. CAB ISFI exhibited high concentrations in vaginal, cervical, and rectal tissues for up to 180 days. Furthermore, depots were easily retrievable up to 180 days post-administration with up to 34% residual CAB and near complete (85%) polymer degradation quantified in depots ex vivo. After depot removal, results demonstrated a median 11-fold decline in CAB plasma concentrations across all doses. Ultimately, this study provided critical PK information for the CAB ISFI formulation that could aid in its future translation to clinical studies.

## 1. Introduction

Globally, there are over 38 million people living with HIV with approximately 1.5 million new infections reported in 2021 [1]. Prior to late 2021, daily oral regimens containing tenofovir and emtricitabine were the only antiretroviral (ARV) prevention methods against HIV. However, effectiveness of daily oral regimens is strictly dependent on high patient adherence [2]. The average adherence to daily HIV pre-exposure prophylaxis (PrEP) regimens is only 70% and suboptimal adherence could lead to the development of drug-resistant virus and lack of protection [2,3]. The development of long-acting technologies for HIV prevention can overcome adherence challenges with daily oral PrEP due to reduced dosing frequency, which can increase patient adherence and reduce stigma [4,5]. 

In December 2021, the FDA approved the first long-acting injectable containing integrase inhibitor cabotegravir (CAB LA) for HIV PrEP given as bimonthly intramuscular injections (200 mg of CAB/mL, 3 mL). Although CAB LA represents a major progression in the HIV prevention landscape, there are a few limitations with the technology, such as large injection volumes (3 mL), injection site reactions [6,7] and need of physician assistance for administration. Furthermore, a long pharmacokinetic (PK) tail is present due to CAB LA’s long half-life (40 days) [8] and the inability to remove the drug once administered to terminate treatment [9]. This PK tail elicits suboptimal but quantifiable levels of CAB in the plasma for over 1 year after the last CAB LA injection, which could cause drug resistant virus or breakthrough infections [9,10]. 

In situ forming implants (ISFIs) may overcome limitations associated with CAB LA due to their ability to achieve ultra-long-acting drug release, compatibility with many drugs including ARVs, biodegradable nature, and ability to be safely removed via a small incision to terminate treatment, if necessary [11,12,13]. ISFIs are comprised of a biodegradable and hydrophobic polymer (e.g., poly(lactic-*co*-glycolic acid) (PLGA)), water-miscible organic solvents (e.g., *N*-methyl-2-pyrrolidone (NMP) or dimethyl sulfoxide (DMSO)), and active pharmaceutical ingredients (APIs). When co-formulated, these components generate a syringeable and homogenous liquid suspension for injection into the subcutaneous or intramuscular space. Upon injection, a phase inversion occurs resulting in a solid depot with the API dispersed within the precipitated polymer matrix [14,15,16]. Drug release from the precipitated implant is driven by diffusion mechanisms or as the polymer degrades over time. PLGA degrades via ester hydrolysis and its degradation rate is largely dependent on molecular weight and its lactic acid to glycolic acid ratio [17]. In our previous ISFI reports, we utilized low molecular weight PLGA, such as 10 kDa or 27 kDa molecular weight, and a lactic acid to glycolic acid ratio of 50:50 to promote a degradation profile of approximately 3 months to ensure that complete polymer degradation occurs prior to subsequent ISFI injections to prevent polymer accumulation [11,13,17,18]. Furthermore, as previously mentioned, ISFIs involve the use of organic solvents within the formulation, such as NMP and DMSO [15,19], which can cause toxicity in large quantities. However, we previously developed an optimized ISFI formulation including NMP and DMSO that showed no local or chronic toxicity when delivered to mice or macaques and solvent quantities were below the median lethal dose (LD_50_) and the no-observable-effect-level (NOEL) when administered to macaques [13,20,21,22,23,24]. 

Specifically, we previously developed a subcutaneously administered ultra-long-acting (>6 months) and well-tolerated PLGA-based ISFI with CAB (500 mg/mL) that demonstrated complete protection in female rhesus macaques after multiple rectal simian–human immunodeficiency virus (SHIV) challenges [13]. Furthermore, we showed that CAB ISFIs can be easily retrieved from macaques 90 days post-injection with a rapid decline in CAB plasma concentrations [13]. Reversibility and the ability to remove implants can aid in preventing a long PK tail, which can occur after discontinuation of treatment. The PK tail consists of low levels of drug remaining in the plasma after treatment termination, which could increase the risk for drug-resistance to the virus [9]. However, when CAB ISFIs were retrieved from macaques, complete CAB elimination in plasma was not achieved after depot removal. This was surprising as a previous study with a dolutegravir (DTG) ISFI (analog of CAB) demonstrated complete elimination of DTG in plasma within 7 days after ISFI removal in humanized mice [11]. Furthermore, due to the ultra-long-acting nature of CAB ISFI, the time to completion of CAB release and polymer degradation after injection and long-term vaginal and rectal tissue PK have yet to be established. Ultimately, detailed characterization of CAB ISFIs is needed for further translation to clinical studies. 

Thus, we sought to perform a comprehensive dose-ranging PK study in mice with CAB ISFI (500 mg/mL) to investigate (1) CAB ISFI time to completion, (2) long-term (180 days) vaginal and rectal tissue PK, (3) CAB PK tail post-depot removal, (4) implant biodegradation, and (5) the effect of dose (50 µL (1215 mg/kg) vs. 100 µL (2430 mg/kg)) and number of injections (100 µL vs. 2× 50 µL) on CAB in vivo drug release (Figure 1). This study allowed us to further characterize the CAB ISFI formulation and obtain crucial PK information that can aid in establishing optimal formulation and administration parameters for clinical translation. 

## 2. Materials and Methods

### 2.1. Materials

Poly(DL-lactide-co-glycolide, PLGA) containing 50:50 lactic acid:glycolic acid was purchased from LACTEL (Birmingham, AL, USA; Cat. No. B6017-1G, Lot# A15-108, molecular weight 10 kDa, inherent viscosity 0.15–0.25 Dl/g). N-methyl-2-pyrrolidone (NMP, USP) was received from ASHLAND (Wilmington, DE, USA; Product Code 851263, 100% NMP). Dimethyl sulfoxide (DMSO, ≥99.7%) was purchased from Sigma-Aldrich (St. Louis, MO, USA; Lot# RNBH5297). HPLC grade acetonitrile (ACN) and water were purchased from Sigma Aldrich (St. Louis, MO, USA). Tetrahydrofuran (THF) was purchased from Sigma Aldrich (St. Louis, Mo, USA; Cat. No. SHBF9530V). High purity (≥99%) CAB was purchased from Selleckchem (Houston, TX, USA; Cat. No. S7766).

### 2.2. High-Performance Liquid Chromatography (HPLC)

A reverse-phase HPLC analysis was performed on an Agilent 1260 HPLC system (Agilent Technologies, Santa Clara, CA, USA) equipped with a Diode Array Detector and an LC pump with autosampler, as previously described [13]. In brief, an Inertsil ODS-3 column (5 μm, 4.6 mm × 150 mm 100 Å, (GL Sciences, Torrance, CA, USA)) maintained at 40 °C was used as the stationary phase, and the mobile phase consisted of 0.1% trifluoroacetic acid in water and ACN. We utilized a flow rate of 1.0 mL/min and a total run time of 25 min per sample. CAB concentration was analyzed at a wavelength of 254 nm with a retention time of 11.4 min. The calibration range was 0.48–250 µg/mL.

### 2.3. Preparation of ISFI Formulation

The preparation of CAB ISFIs has been previously described [13]. In brief, ISFI formulations for mouse studies were prepared under aseptic conditions in a biosafety cabinet. A 50:50 poly(DL-lactide-co-glycolide) (PLGA) (10 kDa molecular weight) was fully dissolved in the solvent (1:1 weight ratio (*w*/*w*) NMP:DMSO) at a ratio of 1:4 *w*/*w*. Solvents and placebo ISFI formulations were sterile filtered (Sterile Puradisc 13 mm Nylon Syringe Filter, 0.2 µm; Cat# 6786-1302) prior to incorporating the drug in the formulation. Cabotegravir (500 mg/mL) was added to the sterile placebo solution producing a stable ISFI suspension and mixed at 37 °C for 48 h. To ensure homogeneity of the drug suspension, sample aliquots were collected from the drug formulation and dissolved in ACN. Drug concentration was quantified by HPLC analysis.

### 2.4. Pharmacokinetic (PK) Studies in Female BALB/c Mice

All in vivo studies were conducted with female BALB/c mice (8–10 weeks, Jackson Laboratory). All experiments involving mice were carried out with an approved protocol by the University of North Carolina Animal Care and Use Committee. Housing conditions consisted of a 12 h/12 h light/dark cycle. The ambient temperature was 68–72 degrees Fahrenheit with 30–70% humidity. All injections were administered subcutaneously with a 19 G needle. All samples were stored at −80 °C until PK analysis. 

#### 2.4.1. Long-Term Plasma PK and Non-Compartmental Analysis 

A 367-day in vivo study was conducted to assess long-term and time-to-completion PK of CAB ISFIs in female BALC/c mice. Mice were subcutaneously injected with 50 µL, 100 µL, or 2× 50 µL of CAB ISFI formulation (n = 6/dose). For this study, 2.5 mL of CAB ISFI formulation was prepared to ensure sufficient volume during injections, and an analytical CAB concentration of 499.1 ± 12.7 mg/mL was determined via HPLC analysis. At 1 h, 1 day, 3 days, 7 days, 30 days, and monthly thereafter, plasma samples were collected longitudinally from mice. It is important to note that CAB plasma concentrations up to 90 days post-injection for mice injected with 50 µL of CAB ISFI have been previously reported [13]. The data in this manuscript have been extended to 367 days post-injection. CAB was extracted from mouse plasma by protein precipitation with methanol containing the stable, isotopically labeled internal standard, 13C,2H2,15N-CAB before analysis by LC-MS/MS. Chromatographic separation was completed on a Waters Atlantis T3 (50 mm × 2.1 mm, 3 µm particle size) analytical column with detection on an AB Sciex API-5000 triple quadruple mass spectrometer. The calibration range was 25–75,000 ng/mL. Calibration standards and quality control samples were within 30% of nominal concentrations.

Area-under-the-curve (AUC) for all injection volumes were calculated using non-compartmental analysis (NCA) in Phoenix WinNonlin (version 8.3) (Certara, L.P., St. Louis, MO, USA). The log-linear trapezoidal method was used to calculate AUCs over the 367-day time course. PK data are represented as median with interquartile range. 

#### 2.4.2. Tissue PK

A 180-day study was conducted to assess the distribution of CAB in vaginal, cervical, and rectal tissue in female BALC/c mice. Mice were subcutaneously injected with 50 µL, 100 µL, or 2× 50 µL of CAB ISFI formulation (n = 3/dose per timepoint). For this study, 6.25 mL of CAB ISFI formulation was prepared to ensure sufficient volume during injections, and an analytical CAB concentration of 499.3 ± 8.6 mg/mL was determined via HPLC analysis. At 7, 30, 60, 90, and 180 days post-injection, vaginal, cervical, and rectal tissues were collected from each mouse, weighed, and stored at −80 °C until PK analysis. Weighed tissues were transferred into Precellys^®^ hard tissue reinforced metal bead kit tubes (Cayman Chemical Company, Ann Arbor, MI, USA) containing 1 mL of 70:30 acetonitrile:water, homogenized, and then centrifuged. Following protein precipitation extraction with the isotopically labeled internal standard, 13C,2H2,15N-CAB, CAB was separated using reverse-phase chromatography via a Waters Atlantis T3 (50 mm × 2.1 mm, 3 µm particle size) analytical column. An AB Sciex API-5000 triple quadruple mass spectrometer was used to detect the analyte and internal standard under positive ion electrospray conditions. Precision and accuracy of the calibration standards and quality control samples were within 15% for the dynamic range of 1.00–4000 ng/mL. Final concentrations were normalized to tissue mass and reported in ng/g.

#### 2.4.3. Depot Removal PK and Assessment of PK Tail

A 180-day in vivo study was conducted to assess plasma concentrations after CAB ISFI removal, residual CAB, and biodegradation of PLGA after ISFI removal in female BALB/c mice. Mice were subcutaneously injected with 50 µL, 100 µL, or 2× 50 µL of CAB ISFI formulation (n = 6/dose per depot removal timepoint). For this study, 9 mL of CAB ISFI formulation was prepared to ensure sufficient volume during injections and an analytical CAB concentration of 497.2 ± 12.6 mg/mL was determined via HPLC analysis. At 1 h, 1 day, 3 days, 7 days, 30 days, 60 days, 90 days, and 180 days (n = 6 mice/timepoint per dose), plasma samples were collected longitudinally from mice. At days 30, 60, 90, and 180 (n = 6 mice/timepoint per dose), CAB ISFI was carefully removed via a small skin incision at the injection site, flash frozen, and stored at −;80 °C until further processing (Section 2.5 and Section 2.6). Plasma samples (n = 6 mice/timepoint per dose) were collected at 1 day, 3 days, 7 days, 14 days, 21 days, and 30 days post-depot removal to assess CAB elimination (i.e., PK tail) over time. Plasma concentrations were analyzed as described in Section 2.4.1 above. 

### 2.5. Residual Drug Quantification

CAB ISFIs were extracted to quantify the amount of CAB remaining in the implant after removal from mice at days 30, 60, 90, and 180 post-administration (n = 3/timepoint per dose). Each implant was massed to assess mass loss over time. To quantify residual drug concentration, implants were dissolved in ACN to extract residual CAB. Residual CAB from the implant was quantified by HPLC analysis. It is important to note that CAB residual drug measurements up to 90 days post-injection for mice injected with 50 µL of CAB ISFI have been previously reported [13]. The data in this manuscript have been extended to 180 days post-injection. 

### 2.6. Gel Permeation Chromatography (GPC)

Gel permeation chromatography (GPC) was conducted as previously described [13]. In brief, GPC was conducted to determine the change in polymer molecular weight over time for ISFIs that were retrieved from mice at days 30, 60, 90, and 180 post-injection (n = 2–3/timepoint per dose). GPC analysis on neat polymer and CAB ISFIs was performed on a Tosoh Biosciences EcoSEC Elite HLC-8420 with a TSKgel GMH-M column (7.8 mm × 30 cm with a pore size of 5 microns). Tetrahydrofuran (THF) was selected as the mobile phase with a flow rate of 0.5 mL/min. Molecular weight was calculated relative to polystyrene standards and analyzed by refractive index detection. It is important to note that polymer molecular weight analysis up to 90 days post-administration for mice injected with 50 µL of CAB ISFI has been previously reported [13]. The data in this manuscript have been extended to 180 days post-injection. 

### 2.7. Statistical Analysis

Statistical analyses were performed in GraphPad Prism version 9.4 (GraphPad Software, Inc., La Jolla, CA, USA). A Kruskal–Wallis one-way ANOVA followed by a Dunn’s multiple comparisons test was performed to analyze differences in drug exposure (AUC) in plasma across doses and number of injections. To assess differences in CAB concentration in vaginal, cervical, and rectal tissues across doses and number of injections, a two-way ANOVA and Tukey’s multiple comparisons test were conducted with respect to dose and timepoint. A *p* value of <0.05 (95% confidence level) was deemed significant.

## 3. Results and Discussion

### 3.1. Dose-Ranging Ultra-Long-Acting PK

Dose-ranging ultra-long-acting PK studies in female BALB/c mice were conducted over 367 days to assess the duration of complete CAB release across different injection volumes and doses (50 µL (1215 mg/kg), 100 µL (2430 mg/kg), and 2× 50 µL (2430 mg/kg)). Mice received a subcutaneous injection of 50 µL, 100 µL, or 2× 50 µL (n = 6 mice/dose) of CAB ISFI (500 mg/mL). CAB plasma concentrations for each dose were analyzed and plotted over 367 days (Figure 2). Mice elicited average plasma concentrations of CAB well above the 4× protein adjusted 90% inhibitory concentration (4× PA-IC_90_, 664 ng/mL) for 11 months when injected with 50 µL of CAB ISFI and 12 months when injected with 100 µL or 2× 50 µL of CAB ISFI. Due to the longer than expected duration of the CAB ISFI, time to completion was not determined and this study was terminated at 367 days. However, mice that were injected with 50 µL of CAB ISFI elicited a 19-fold decline in CAB plasma concentrations between day 330 and 367, exhibiting average concentrations near the 1× PA-IC_90_ (166 ng/mL). We could expect the 100 µL and 2× 50 µL of CAB ISFI groups to follow a similar trend if the study continued for an additional couple of months. 

Furthermore, the results demonstrated proportionality between dose (1215 mg/kg (50 µL) vs. 2430 mg/kg (100 µL)) and overall CAB exposure (AUC) (Table 1). Additionally, AUCs were not significantly different between the number of injections with the same dose (1 × 100 µL vs. 2× 50 µL) (*p* = 0.3372) (Table 1). Ultimately, these data suggest that (1) CAB ISFI can provide CAB plasma concentrations above the protective benchmark (4× PA-IC_90_) for 11–12 months reducing dosing frequency compared with marketed products and (2) CAB ISFI dose is approximately proportional to overall drug exposure and there is no difference in drug exposure when administering the same dose with multiple injections. Knowing these characteristics of the formulation, if CAB ISFI were to fall below the protective benchmark in large animal or human studies, an increase in injection volume could provide the ability to reach protective plasma levels due to proportionality with drug exposure. 

### 3.2. Ultra-Long-Acting Genital Tissue PK 

CAB concentrations were measured in vaginal, cervical, and rectal tissues in female BALB/c mice up to 180 days post-CAB ISFI injection. These genital tissues represent main entry points for HIV infection; thus, it is crucial to ensure CAB concentrations are high in these target tissues to provide protection against HIV infection. Figure 3A–C show CAB concentrations in cervical, vaginal, and rectal tissues for 180 days, respectively. CAB concentrations were high in all tissues with slightly greater concentrations in cervical and vaginal tissues compared with rectal tissues. CAB concentrations increased across all tissues for 30 days and remained stable until 180 days. Furthermore, tissue-to-plasma ratios across doses remained stable over the study period (180 days) for vaginal, cervical, and rectal tissue (Table 2). Additionally, there was no significant difference between CAB tissue concentrations across doses (*p* > 0.05). 

### 3.3. CAB ISFI Removal and Assessment of PK Tail

Currently available CAB LA for HIV prevention demonstrated a PK tail for at least 15 months due to CAB’s long half-life (40 days) and inability to be removed to terminate treatment [9]. Investigating the PK tail is of upmost importance as suboptimal and detectable levels of CAB that remain in the plasma can cause drug-resistant virus or breakthrough infections. Since ISFIs are administered subcutaneously and form a depot under the skin, treatment can easily be terminated via a small skin incision at the injection site. However, it is important to note that CAB ISFIs are biodegradable and only need to be removed if necessary (i.e., allergic reaction, adverse events, desire to discontinue treatment, no longer at risk for HIV). In our previous work, we assessed the PK tail by removing CAB ISFIs from two rhesus macaques. It was found that 1 macaque demonstrated a 100-fold decline in CAB plasma concentration and fell below the limit of quantification within 2 weeks post-removal [13]. The other macaque only exhibited a 10-fold decline in CAB concentrations within 2 weeks after ISFI removal. Thus, we sought to further investigate this phenomenon by removing CAB ISFI from mice at days 30, 60, 90, and 180 post-administration and quantifying CAB plasma concentration 30 days post-ISFI removal (Figure 4). 

Results showed that plasma CAB levels exhibited a significant decrease after implant removal across doses (Figure 4A–D). Mean fold decline of CAB plasma concentrations across doses are illustrated in Table 3. 

When CAB ISFIs were removed 30 and 60 days post-administration, mice exhibited a rapid decline in CAB plasma concentrations within 7 days post-removal, but CAB concentrations reached a plateau between day 7 and day 30 post-removal (Figure 4A,B). Interestingly, when CAB ISFIs were removed 90 and 180 days post-administration, there was a continuous decline in CAB plasma concentrations for 30 days post-removal (Figure 4C,D), however, complete CAB elimination was not achieved. From this, we hypothesize that there may be an accumulation of CAB in the tissues, particularly in the subcutaneous tissue at the injection site, due to the high dose of CAB administered (1215 mg/kg (50 μL) or 2430 mg/kg (100 μL or 2× 50 μL)). This potential reservoir may explain why CAB plasma levels were not completely eliminated after 30 days post-ISFI removal. Future studies will include assessing the complete biodistribution of the CAB ISFI after injection, particularly the CAB concentration at the subcutaneous tissue injection site to determine the presence of a drug reservoir. Additionally, we plan to determine the exact duration of the PK tail by quantifying CAB concentrations in plasma until they are no longer detectable. Ultimately, CAB ISFI removal resulted in a rapid and considerable decline in CAB concentration in plasma, but complete elimination was not achieved. 

### 3.4. CAB ISFI Retrievability, Residual CAB, and Polymer Biodegradation

To assess CAB ISFI retrievability over time, we removed CAB ISFIs from mice via a small skin incision at the injection site at days 30, 60, 90, and 180 post-administration for each dose. Six mice per timepoint per dose were evaluated. However, 1 mouse that received a 100 µL injection for day 60 depot removal and 1 mouse that received a 2× 50 µL injection for day 180 depot removal were removed from the study prior to implant retrieval due to health concerns. ISFIs across doses were successfully removed from mice with no fibrotic tissue surrounding the depot (Figure 5A). For 1 of the mice, the depot retrieved at day 30 post-administration was significantly smaller (the mouse received a 100 µL injection), which was likely due to incomplete depot removal. There was a mass loss of 36.7 ± 5.2%, 44.5 ± 2.4%, 52.0 ± 6.5%, and 62.2 ± 11.6% relative to depot mass at day 0 across doses when CAB ISFIs were removed at days 30, 60, 90, and 180 post-administration, respectively (Figure 5B). 

Depots were further processed to evaluate residual CAB and polymer degradation. After 180 days in mice, there was up to 34% CAB remaining (Figure 6A and Table 4) and approximately 15% polymer remaining (Figure 6B and Table 5). Notably, residual CAB and PLGA molecular weight over time were fairly consistent across doses, thus suggesting proportionality between dose and CAB release rate and polymer degradation. Furthermore, the depots were still intact and easily retrievable up to 180 days post-administration despite PLGA being almost completely degraded (85% PLGA molecular weight decrease). This is likely attributed to the high CAB content within the ISFI (41.2% CAB and 11.8% PLGA) which greatly contributes to the depot’s structural integrity. 

Since only 23–34% of CAB remained after 180 days, it would be unlikely for CAB ISFI to produce protective concentrations for an additional 6 months. However, as previously seen in Figure 2, high concentrations of CAB in plasma were sustained for 330–367 days. This continues to suggest the possibility of a drug reservoir in tissues, particularly in the subcutaneous tissue at the injection site contributing to sustained plasma concentrations even if a negligible amount of CAB remains in the implant. Additionally, commercially available CAB LA is known to have a long half-life (40 days) [8], which could also attribute to drug accumulation and promote long-lasting CAB plasma concentrations. Therefore, it is critical to perform a complete biodistribution study in the future. Ultimately, these results further characterize the formulation defining CAB release from the implant and polymer biodegradation up to 180 days in vivo. 

## 4. Conclusions

We performed a comprehensive dose-ranging PK study of CAB ISFIs in mice and assessed the effect of dose and number of injections on CAB ISFI PK, time to completion of CAB release and polymer degradation, long-term genital tissue PK, and CAB PK tail post-depot removal. After a single administration of CAB ISFI, plasma concentrations were maintained above the 4× PA-IC_90_ for 11–12 months, owing to its ultra-long-acting capability. Additionally, proportionality between CAB ISFI dose and overall drug exposure was observed. Furthermore, we demonstrated high CAB concentrations in vaginal, cervical, and rectal tissues for 180 days with no significant differences between doses or number of injections. When assessing the CAB PK tail, a significant decline in CAB plasma concentrations was observed within 30 days after depot removal, but complete CAB elimination was not achieved, which may be due to the presence of a CAB reservoir in the subcutaneous tissue. After 180 days post-injection, CAB ISFIs were easily retrievable with up to 34% CAB remaining and an 15% polymer remaining. These results confirm CAB ISFI’s ultra-long-acting ability with high concentrations in plasma and target genital tissues, which are common sites of HIV infection. Ultimately, this study further characterized the PK of the CAB ISFI formulation and can assist with determining optimal administration parameters for clinical translation. In future, we plan to perform a complete biodistribution study in mice and similar studies should be performed in macaques to confirm results with clinically relevant injection volumes and doses. 

## 5. Patents

SRB, ICY, and RS are inventors on a patent describing the Injectable Polymer-Based Drug Formulation for Ultra-Long-Acting Drug Delivery.

## Figures and Tables

**Figure 1 pharmaceutics-15-01487-f001:**
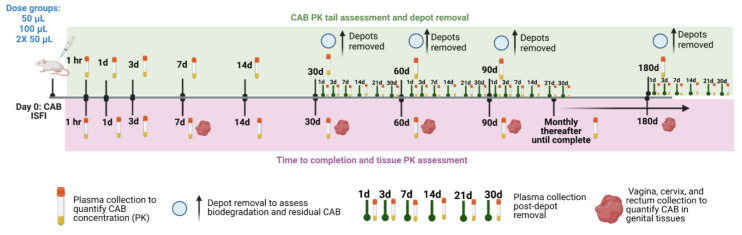
Study plan of dose-ranging CAB ISFI PK. Female BALB/c mice were subcutaneously injected with 50 µL (1215 mg/kg), 100 µL (2430 mg/kg), or 2× 50 µL (2430 mg/kg) of CAB ISFI to assess CAB PK tail and implant biodegradation (n = 6 mice/dose), CAB ISFI time to completion (n = 6 mice/dose), and long-term genital tissue (vagina, cervix, and rectum) PK (n = 3 mice/dose). Figure created with Biorender.com.

**Figure 2 pharmaceutics-15-01487-f002:**
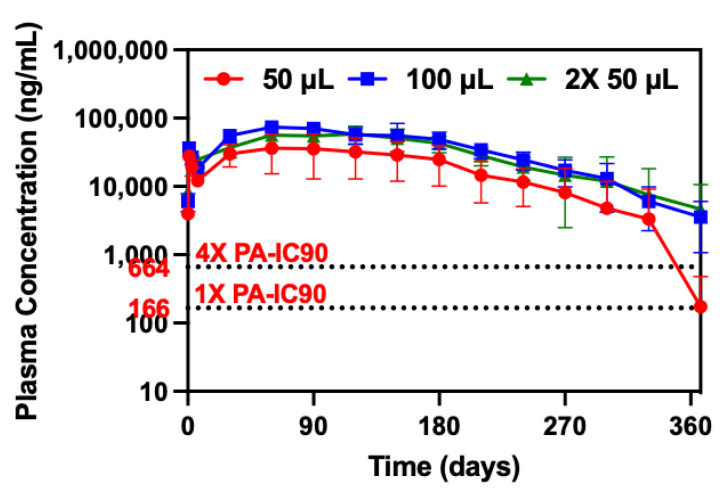
Ultra-long-acting CAB ISFI PK and overall drug exposure. Plasma concentrations of CAB after 50 µL, 100 µL, or 2× 50 µL injection of CAB ISFI in female BALB/c mice for 367 days. Data are presented as average ± standard deviation of n = 5–6 mice/timepoint per dose. Individual replicates are shown in Appendix A. CAB plasma concentration data for mice injected with 50 µL of CAB ISFI up to 90 days have been previously reported [13] (Reprinted with permission from Ref. [13]. 2023, Young, IC & Massud, I et al. Nature Communications).

**Figure 3 pharmaceutics-15-01487-f003:**
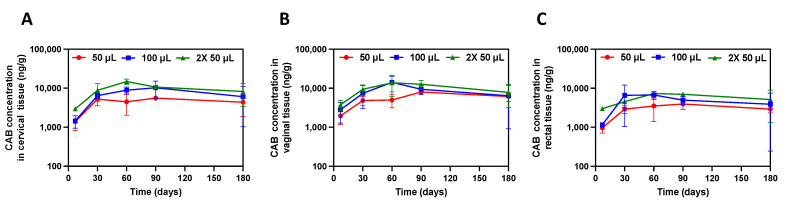
Genital tissue concentrations after CAB ISFI administration. (**A**) CAB concentration in cervical tissue (ng/g). (**B**) CAB concentration in vaginal tissue (ng/g). (**C**) CAB concentration in rectal tissue (ng/g). Data are presented as average ± standard deviation of n = 3 mice/timepoint per dose. Individual replicates are shown in Appendix A.

**Figure 4 pharmaceutics-15-01487-f004:**
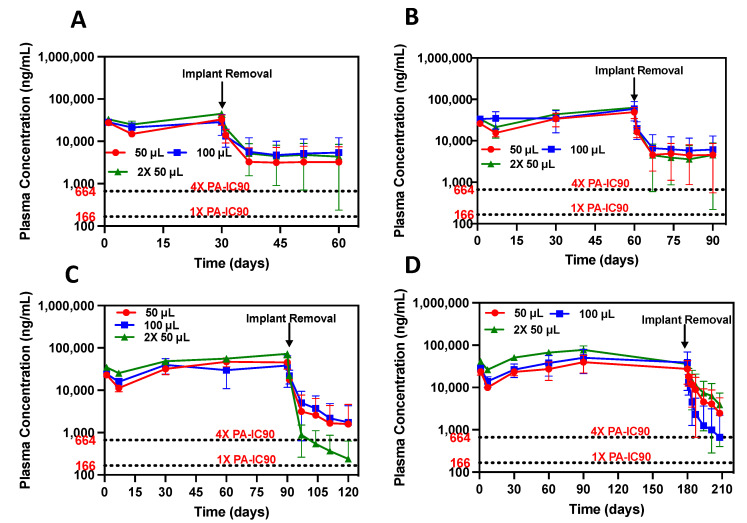
CAB ISFI removal and PK tail assessment. (**A**–**D**) CAB plasma concentration after ISFI removal at 30, 60, 90, and 180 days post-injection, respectively. Data are presented as average ± standard deviation for n = 6 mice/timepoint per dose. Individual replicates shown in Appendix A.

**Figure 5 pharmaceutics-15-01487-f005:**
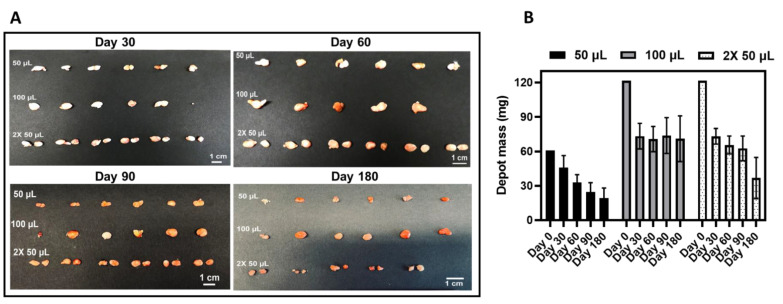
CAB ISFI removal and change in depot mass over 180 days. (**A**) Images of CAB ISFIs retrieved across doses after 30, 60, 90, and 180 days post-administration. (**B**) CAB ISFI masses after retrieval at 30, 60, 90, and 180 days post-administration (data are represented as average ± standard deviation for n = 5 or 6 mice/timepoint) compared with initial ISFI mass (day 0) of 60.75 mg (50 µL injection volume) or 121.5 mg (100 µL total injection volume). Day 0 masses were determined based on the density of the formulation (1.215 g/mL). Depot images and masses of CAB ISFI for mice injected with 50 µL of CAB ISFI up to 90 days have been previously reported [13]. (Reprinted with permission from Ref. [13]. 2023, Young, IC & Massud, I et al. Nature Communications).

**Figure 6 pharmaceutics-15-01487-f006:**
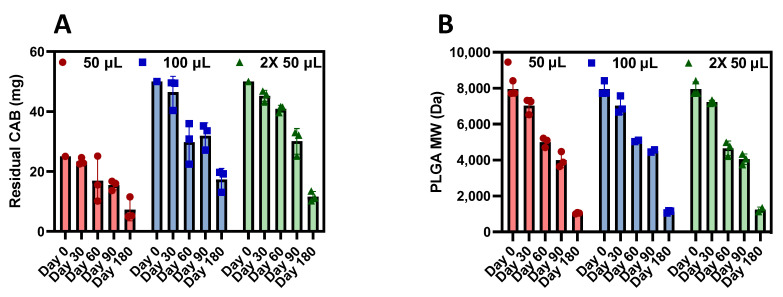
Residual CAB and PLGA biodegradation after CAB ISFI removal. (**A**) Residual CAB from ISFIs 30, 60, 90, and 180 days post-administration in BALB/c mice across injection volumes (average ± standard deviation of n = 3 samples per timepoint per injection volume). The initial CAB dose was 25 mg of CAB in 50 µL injection or 50 mg of CAB in 100 µL total injection. (**B**) PLGA molecular weight from CAB ISFIs 30, 60, 90, and 180 days post-administration in mice (average ± standard deviation of n = 2–3 samples per timepoint per injection volume) compared with neat PLGA (10 kDa). Residual CAB and polymer molecular weight data up to 90 days for mice injected with 50 µL of CAB ISFI have been previously reported [13]. (Reprinted with permission from Ref. [13]. 2023, Young, IC & Massud, I et al. Nature Communications).

**Table 1 pharmaceutics-15-01487-t001:** CAB ISFI median AUCs over 367 days for all doses and injections.

Group	Dose (mg)	Median AUC_0→367_ (day × ng/mL)	25% Percentile	75% Percentile
50 µL	25	7,222,125	4,375,526	9,306,997
100 µL	50	14,731,084	13,075,480	15,226,388
2× 50 µL	50	12,634,860	11,474,641	12,906,369

**Table 2 pharmaceutics-15-01487-t002:** Average tissue-to-plasma ratios after CAB ISFI injection in female BALB/c mice.

Group	Tissue	Day 7	Day 30	Day 60	Day 90	Day 180
50 µL	Cervical	0.114 ± 0.05	0.175 ± 0.06	0.123 ± 0.07	0.157 ± 0.02	0.177 ± 0.10
	Vaginal	0.160 ± 0.06	0.162 ± 0.04	0.137 ± 0.05	0.226 ± 0.03	0.247 ± 0.07
	Rectal	0.079 ± 0.02	0.097 ± 0.02	0.097 ± 0.06	0.111 ± 0.03	0.118 ± 0.02
100 µL	Cervical	0.078 ± 0.03	0.117 ± 0.02	0.120 ± 0.02	0.147 ± 0.07	0.124 ± 10
	Vaginal	0.152 ± 0.08	0.133 ± 0.08	0.192 ± 0.08	0.134 ± 0.03	0.132 ± 0.11
	Rectal	0.063 ± 0.01	0.119 ± 0.10	0.091 ± 0.02	0.071 ± 0.02	0.080 ± 0.07
2× 50 µL	Cervical	0.119 ± 0.01	0.240 ± 0.11	0.263 ± 0.04	0.195 ± 0.01	0.193 ± 0.11
	Vaginal	0.153 ± 0.04	0.255 ± 0.08	0.244 ± 0.14	0.232 ± 0.06	0.183 ± 0.11
	Rectal	0.121 ± 0.01	0.123 ± 0.03	0.129 ± 0.02	0.128 ± 0.01	0.120 ± 0.09

**Table 3 pharmaceutics-15-01487-t003:** Average fold decline in CAB plasma concentrations after 30 days post-ISFI removal.

Group	30-Day Implant Removal	60-Day Implant Removal	90-Day Implant Removal	180-Day Implant Removal
50 µL	10.11 ± 2.09	10.70 ± 3.55	28.68 ± 4.61	11.08 ± 4.81
100 µL	5.23 ± 2.16	9.61 ± 4.19	22.06 ± 10.44	58.37 ± 19.54
2× 50 µL	10.29 ± 1.87	13.92 ± 2.50	303.79 ± 36.41	10.81 ± 2.14

**Table 4 pharmaceutics-15-01487-t004:** Summary of % CAB remaining after depot removal over time across injection volumes.

Injection Volume	% CAB Remaining			
	30 Days	60 Days	90 Days	180 Days
50 µL	93.2 ± 4.5%	67.6 ± 30.4%	61.7 ± 6.5%	23.2 ± 3.4%
100 µL	92.3 ± 10.5%	59.5 ± 13.6%	63.8 ± 8.5%	34.6 ± 7.4%
2× 50 µL	90.4 ± 3.6%	81.7 ± 2.3%	60.1 ± 8.5%	29.0 ± 14.8%

**Table 5 pharmaceutics-15-01487-t005:** Summary of % PLGA molecular weight (MW) decrease over time across injection volumes.

Injection Volume	% PLGA MW Decrease			
	30 Days	60 Days	90 Days	180 Days
50 µL	11.6 ± 5.6%	37.1 ± 3.4%	49.7 ± 5.4%	86.9 ± 0.43%
100 µL	11.6 ± 5.9%	36.3 ± 0.7%	43.3 ± 1.1%	85.8 ± 0.83%
2× 50 µL	9.1 ± 1.3%	41.5 ± 5.1%	49.1 ± 3.7%	84.5 ± 1.89%

## Data Availability

The data presented in this study are available upon request from the corresponding author.

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
