# Peer review of "Dose-Ranging Plasma and Genital Tissue Pharmacokinetics and Biodegradation of Ultra-Long-Acting Cabotegravir In Situ Forming Implant"

_pharmaceutics, 2023, doi:10.3390/pharmaceutics15051487_

Round 1

Reviewer 1 Report

The authors in the paper report the in-vivo evaluation of a subcutaneous polymeric implant for delivering CAB antiviral in a sustained-long term manner thereby having potential implications in the proactive prevention of STD/viral infections. The manuscript is well written, easy to read and understand. I believe the research has significant importance to the field of reproductive healthcare. I have a few comments/suggestions and hope the authors would be able to address them.

Major comments

1)  Can the authors explain what 50:50 PLGA is? What is the “chapter 2” mentioned in section 2.4.1?

2)  The Depot is made of PLGA solids that precipitate out with time and encapsulates the drugs. What does the solids look like? Are these a mixture of irregular solid fragments? Thin films at the site of injection? Can CAB and the polymer be differentiated visually?

3)    In table 2, is the change at 90 days 303.79-fold?

4)  What would the release profile be for a CAB only formulation in NMP: DMSO if given subcutaneously? Since CAB also contributes to structural integrity of the depot mass and is sparingly soluble in aqueous buffers.

5)  Please include references and a brief description of PLGA degradation rate and mechanisms.

6) Figure 6A and C, in comparison to figure 5, the 100ul formulation, any specific reason as to decreasing CAB content and PLGA length, but consistent depot mass?

Minor comments

I encourage the authors to proofread the comments and improve the readability of the paper, for e.g. line 230- “Instead of to” and expand abbreviation at least once in the paper (SHIV, NMP, etc.).

Author Response

Dear reviewer,

We have attached a detailed point-by-point response to your comments and revised the manuscript to address your comments. 

We would like to thank you for your valuable suggestions and comments which have improved our manuscript. 

Regards,

Rahima

Reviewer 2 Report

The authors have described  studies in a mouse model to further characterise pharmacokinetics and tissue distribution of cabotegravir dosed as an in situ forming implant.  These implants are a known approach to long acting depot formulation administration, established in veterinary medicine and  of interest in creating novel and clinically valuable human therapeutics.

The paper is well written and suitable for further consideration for publication in Pharmaceutics, subject to the authors addressing a few comments given below.

Page 2 lines 51 - 61: please add brief comment as to safety and regulatory acceptance of the organic solvents  for use in human medicines.

Page 2 lines 67-69:  the extended elimination tail post device removal in the macacque model is of itself not a surprise given the significant difference in half life between cabotegravir and dolutegravir.  Authors should rephrase this more clearly.  I think they found marked differences between animals studied which were not explainable by the intrinsic long half life of cabotegravir. Please explain the concern about the pharmacokinetics post device removal more precisely. Agree that further characterisation, as undertaken in this work, is required.

Page 3 line 115: as the drug is presumed to be in suspension and particle size might influence drug release profile, how was particle size controlled for the drug product preparation?

Page 3 line section 2.4.1: the injected product is a suspension of the drug?  Ff so how  was uniformity of dose confirmed for the  small volumes administered.

Page 4 lines 182-183: how was quantitative removal of the device assured?  Depending how the liquid spreads before the implant forms in situ it may be there are extended filaments of polymer hard to see and easily remove - is this a cause of the variable elimination kinetics after supposed removal? Would adding colouring/fluorescing agent to the devices in future studies help with this?  You hypothesise later - page 8 lines 301 - 305 about local tissue reservoir development due to high dose driving this when noted in the mouse model- is there additional drug physicochemical information to support this?

Author Response

Dear reviewer,

We have attached a detailed point-by-point response to your comments and revised the manuscript to address your comments. 

We would like to thank you for your valuable suggestions and recommentdations which have improved our manuscript. 

Regards,

Rahima

Reviewer 3 Report

The authors have investigated the pharmacokinetic properties of carbotegavir (CAB) ultra-long-acting injectable, biodegradable, and removable in situ forming implant (ISFI)  in animal (mouse) studies. For the study, a formulation was prepared and then plasma concentrations and drug accumulation in different tissues were observed after administering different doses. After 6 months, the implant was removed and the biodegradability of the polymer and residual drug content were investigated. The manuscript is basically well structured, and the figures and tables are informative.

Some critical comments:

- Exactly how large a batch of the formulation was produced (part 2.3)? What was the particle size of the active substance in the suspension prepared? Furthermore, the authors write that the drug content was determined in order to test homogeneity (page 3, lines 117-118). However, the results of this are missing from the manuscript.

- In chapter 2.4, the number of mice used for the measurement is not given.

- The use of commas in the table for Fig 2 is confusing

- No standard deviation is given for the values in the first and second tables.

- The reference list could be extended with the drug delivery systems.

Author Response

(The authors gave the same response as above.)

Round 2

Reviewer 1 Report

I thank the authors for responding to my comments.

The quality of English language is fine.

Reviewer 3 Report

Dear Authors, dear Editor,

I accept the answers. No further questions on the manuscript